# A Lightweight Multi-Stage Visual Detection Approach for Complex Traffic Scenes

**DOI:** 10.3390/s25165014

**Published:** 2025-08-13

**Authors:** Xuanyi Zhao, Xiaohan Dou, Jihong Zheng, Gengpei Zhang

**Affiliations:** 1School of Electronic Information and Electrical Engineering, Yangtze University, Jingzhou 434023, China; 2School of Urban Construction, Yangtze University, Jingzhou 434023, China

**Keywords:** intelligent traffic surveillance, object detection, image dehazing, low-light enhancement

## Abstract

In complex traffic environments, image degradation due to adverse factors such as haze, low illumination, and occlusion significantly compromises the performance of object detection systems in recognizing vehicles and pedestrians. To address these challenges, this paper proposes a robust visual detection framework that integrates multi-stage image enhancement with a lightweight detection architecture. Specifically, an image preprocessing module incorporating ConvIR and CIDNet is designed to perform defogging and illumination enhancement, thereby substantially improving the perceptual quality of degraded inputs. Furthermore, a novel enhancement strategy based on the Horizontal/Vertical-Intensity color space is introduced to decouple brightness and chromaticity modeling, effectively enhancing structural details and visual consistency in low-light regions. In the detection phase, a lightweight state-space modeling network, Mamba-Driven Lightweight Detection Network with RT-DETR Decoding, is proposed for object detection in complex traffic scenes. This architecture integrates VSSBlock and XSSBlock modules to enhance detection performance, particularly for multi-scale and occluded targets. Additionally, a VisionClueMerge module is incorporated to strengthen the perception of edge structures by effectively fusing multi-scale spatial features. Experimental evaluations on traffic surveillance datasets demonstrate that the proposed method surpasses the mainstream YOLOv12s model in terms of mAP@50–90, achieving a performance gain of approximately 1.0 percentage point (from 0.759 to 0.769). While ensuring competitive detection accuracy, the model exhibits reduced parameter complexity and computational overhead, thereby demonstrating superior deployment adaptability and robustness. This framework offers a practical and effective solution for object detection in intelligent transportation systems operating under visually challenging conditions.

## 1. Introduction

With the rapid pace of urbanization and the continuous expansion of transportation infrastructures, modern urban traffic environments are becoming increasingly complex. This growing complexity presents unprecedented challenges for traffic safety monitoring and management. Traditional manual surveillance and rule-based detection approaches have proven inadequate in addressing high-density, multi-object scenarios and adverse weather conditions, falling short of the dual requirements for real-time performance and detection accuracy.

In recent years, computer vision technologies [1,2]—particularly those based on deep learning—have shown significant promise in transportation-related applications. By leveraging convolutional neural networks and real-time image analysis techniques, intelligent object detection systems have demonstrated remarkable robustness in adverse conditions such as rain, snow, and fog. These systems enable rapid identification and precise tracking of key traffic participants, including vehicles and pedestrians. Such technological advancements offer crucial support for the development of efficient, reliable, and intelligent transportation systems, with the potential to substantially enhance road safety, alleviate traffic congestion, and facilitate automated traffic management.

Several recent studies have contributed to advancing this field. Kenk et al. [3] introduced the Vehicle Detection in Adverse Weather Nature dataset, which comprises real-world traffic imagery acquired under adverse meteorological conditions. Luo et al. [4] proposed a multi-scale detection framework based on Faster R-CNN, optimized using neural architecture search. Yao et al. [5] improved YOLOv3 performance by incorporating a larger convolutional kernel and employing k-means++ clustering. Chen et al. [6] developed an edge-based traffic flow detection scheme, while Li et al. [7] addressed domain shift challenges by proposing a context-aware detection framework using Faster R-CNN. Zhang et al. [8] designed a multi-sensor system that integrates millimeter-wave radar and camera data, effectively mitigating performance degradation caused by lighting and weather variability. Additionally, Azimov et al. [9] developed a real-time vehicle detection and tracking system using a large-scale annotated highway dataset, and Rajalakshmi et al. [10] combined deep learning with frame differencing to classify vehicle types for traffic analysis applications.

Despite the significant advancements achieved in recent studies, there remains a critical gap in the development of lightweight solutions that can seamlessly integrate detection accuracy and robustness for complex traffic scenarios under extreme weather conditions. This study is specifically designed to tackle the key challenges associated with intelligent vehicle and pedestrian detection in complex urban traffic environments, including image degradation, multi-scale occlusion, and fine-grained target variation. To this end, we propose an innovative and robust visual detection framework that incorporates an enhanced You Only Look Once (YOLO)-based architecture and a multi-stage image enhancement process. By integrating these components, our framework significantly enhances adaptability and detection accuracy across the entire detection pipeline, from image preprocessing and semantic modeling to detection inference, thereby rendering it highly suitable for real-world deployments. The principal contributions of this work are summarized as follows:Development of a unified image preprocessing module that combines haze removal and low-light enhancement based on ConvIR and CIDNet networks, respectively. This module restores image quality under adverse conditions such as fog and nighttime scenarios, thereby improving the recognizability and structural integrity of the input data.Introduction of a novel illumination enhancement strategy that integrates frequency-domain modeling and luminance residual compensation. For the first time in traffic scene applications, the HVI color space is employed to achieve localized dynamic enhancement in dark regions while maintaining color fidelity, thus addressing the challenge of weak target perception under low-light conditions.Design of a lightweight yet highly robust detection backbone, termed MDR-YOLO (A Mamba-Driven Lightweight Detection Network with RT-DETR Decoding for Complex Traffic Scenes), which integrates VSSBlock for state-space modeling and XSSBlock for stacked feature fusion. This design significantly enhances the network’s representational capacity and localization accuracy for small-scale and distant targets in complex traffic environments.

The remainder of this paper is organized as follows: Section 2 provides a detailed description of the datasets and methods employed. Section 3 presents the experimental results. Section 4 offers an in-depth discussion based on the experimental findings. The conclusions are drawn in Section 5.

## 2. Materials and Methods

To effectively address image degradation challenges—such as haze, low illumination, and structural blurring—that frequently occur in complex urban traffic environments, this study proposes a comprehensive visual perception framework that integrates multi-stage image enhancement with a lightweight detection architecture. The overall workflow of the proposed system is illustrated in Figure 1.

The system begins by ingesting traffic surveillance images and proceeds to a preprocessing stage that comprises two critical modules: image dehazing and illumination enhancement. These modules are designed to suppress haze-induced artifacts, restore edge sharpness, and enhance visual clarity, particularly in low-light regions. By improving the structural integrity and perceptual quality of input images, the preprocessing stage ensures more effective downstream feature extraction.

Subsequently, the enhanced images are passed to the detection stage, where deep feature extraction is performed using an improved backbone network—MDR-YOLO. This network integrates the VSSBlock and VisionClueMerge modules to enhance the model’s capability in capturing multi-scale and spatially structured semantic information. Furthermore, the XSSBlock module facilitates refined multi-scale feature fusion, while the RTDETRDecoder is employed to carry out accurate boundary regression and categorical classification.

Through the synergistic optimization of front-end image enhancement and back-end structural modeling, the proposed framework significantly improves both detection accuracy and robustness under degraded visual conditions. This methodology is particularly effective for vehicle and pedestrian perception in challenging scenarios such as nighttime monitoring, occluded intersections, and visually complex urban environments.

### 2.1. Data Acquisition

To evaluate the efficacy and practicality of our approach for traffic detection under adverse weather conditions, this study employed the Traffic Detection Project dataset provided by Kaggle, which primarily consists of images sourced from Turkey. This dataset encompasses images captured under diverse weather conditions, lighting scenarios, and traffic contexts, rendering it highly suitable for real-world applications.

The dataset comprises 6614 traffic scene images with varying resolutions. For the purpose of this study, the dataset was modified to include four categories of detection targets: car, person, bicycle, and motorbike, in order to better align with practical traffic detection requirements. The dataset was subsequently partitioned into training, testing, and validation sets in a ratio of 8:1:1.

To validate the effectiveness of our image dehazing and illumination enhancement methods, the images in the test set were processed to simulate hazy and low-light conditions. This paper synthesizes physically consistent fogging degradation of fog-free images based on the classic atmospheric scattering model. This model represents the radiance of each pixel in the image as a weighted combination of the clear image and atmospheric illumination, specifically expressed as(1)$$I\left(x\right)=J\left(x\right)\cdot t\left(x\right)+A\cdot \left(1-t\left(x\right)\right)$$

*I*(*x*) represents the fogged image after synthesis, *J*(*x*) represents the original clear image, A is the global atmospheric illumination vector, and *t*(*x*) represents the transmittance, which is calculated as(2)$$t\left(x\right)=e^{-\beta d\left(X\right)}$$

The transmittance depends on the depth value *d*(*X*) of the pixel and the fog concentration control parameter *β*, which is used to simulate different levels of fog intensity.

By adjusting the fog concentration parameter *β*, this paper constructs multiple levels of fogged images to enhance the robustness of the model under different degrees of degradation. Each clear image can generate several fogged images and their corresponding depth maps, forming standard training pairs for network training in the supervised learning framework. This data synthesis strategy avoids the high cost of annotating real fogged images and improves the adaptability and generalization performance of the ConvIR model to variable weather conditions, providing a stable and reliable image input foundation for subsequent target detection tasks.

In this study, Gamma Correction was employed to darken normally exposed images in a physically consistent manner, thereby simulating the low-illumination degradation characteristic of real-world low-light scenes. The mathematical formulation is defined as(3)s=c·r^γ
where *r* represents the input grayscale value, *s* is the output grayscale value, *c* is a normalization constant, and *γ* is the control factor: *γ* > 1 results in an overall darkening of the image, while *γ* < 1 leads to an overall brightening. To closely mimic real low-light scenarios, this study uniformly sampled *γ* within the interval [1.5, 4.0] at a step size of 0.25. Each normally illuminated image was thus processed to generate multiple low-illumination images with varying degrees of darkness, along with their corresponding *γ* parameter labels. These pairs formed the standard training pairs required for supervised learning.

This strategy effectively circumvents the prohibitive costs associated with the acquisition of real nighttime scene data and pixel-wise illumination labeling. Moreover, it enhances the network’s robustness and generalization ability across different levels of low-light degradation, thereby providing a stable and reliable basis for subsequent illumination enhancement and object detection tasks under low-illumination conditions.

### 2.2. Image Dehazing

Under adverse weather conditions, traffic surveillance images often exhibit low contrast, structural blurring, and edge attenuation, all of which significantly degrade the accuracy of key object detection tasks such as pedestrian and vehicle recognition.

To address these issues, various dehazing approaches have been proposed in the literature. Matelin et al. [11] focused on fog and noise removal from single images. Fang et al. [12] introduced a fast variational method for estimating the transmission map using an adaptive window strategy based on the dark channel prior. Wang et al. [13] developed a physically grounded algorithm that combines multi-scale retinal processing with color restoration techniques to enhance dehazing performance. Shin et al. [14] proposed a convolutional neural architecture capable of jointly estimating environmental illumination and the transmission map, with specific emphasis on underwater image enhancement. Yang et al. [15] presented Proximal Dehaze-Net, which embeds fog-related priors within a deep learning framework. Engin et al. [16] introduced “Cycle Dehazing”, a CycleGAN-based model that enables unpaired image dehazing. Liu et al. [17] proposed the Trident Dehazing Network (TDN), employing a coarse-to-fine structure to more effectively handle regions of dense fog and variable fog concentration. Dong et al. [18] incorporated physical modeling to guide feature extraction in a physics-inspired dehazing network. Ura et al. [19] proposed LD-Net, a computationally efficient model that simultaneously estimates transmission and atmospheric illumination while preserving effective dehazing capability.

This paper introduces the ConvIR model as the image preprocessing module at the front end of the detection system. ConvIR is an end-to-end image dehazing network based on convolutional feature interaction, with strong structure preservation and detail reconstruction capabilities, particularly suitable for tasks requiring high-fidelity image restoration in traffic scenarios.

The ConvIR network adopts a multi-branch convolution decoder combined with attention mechanism and frequency domain enhancement path in its structure. In the encoding stage, convolution layers and channel attention modules are stacked to extract low-frequency illumination and high-frequency texture information, focusing on modeling the edge loss and brightness degradation in the fogged area. In the decoding stage, multi-scale feature fusion and bidirectional residual enhancement mechanisms are introduced to achieve layer-by-layer restoration of semantic information and detailed structures. To further enhance the structural perception ability, the ConvIR introduces a frequency domain modeling module, which performs Fourier transformation on intermediate features:(4)$$I_{out}=F_{r}\left(I_{in}\right)+I_{in}$$(5)$$F^{\prime }\left(I\right)=F^{-1}\left(H\left(f\right)\cdot F\left(I\right)\right)$$

*F* and $F^{-1}$ represent the Fourier transform and its inverse operation, respectively. *H*(*f*) is the frequency domain enhancement kernel function, which is used to enhance the global brightness structure and texture contrast of the image.

The network introduces a brightness-guided residual enhancement mechanism in the reconstruction path to perform directional compensation for the uneven brightness and blurred edges of the image:(6)$$\hat{J}=I+R\left(I,L\right)$$

*R*(*∙*) represents the residual mapping function constructed by the image and the luminance guidance map. The entire network training adopts the joint optimization of L1 reconstruction loss and SSIM structural similarity, balancing pixel accuracy and structural fidelity, to enhance the naturalness and perceptual quality of the dehazed images. In traffic scenarios, this illumination enhancement module can significantly improve the local contrast and structural clarity of target areas such as zebra crossings, vehicle outlines, and pedestrian edges, providing a more robust input image basis for subsequent target detectors.

### 2.3. Image Illumination Enhancement

In complex urban traffic environments—particularly during nighttime or in low-light scenarios such as tunnel entrances—images captured by surveillance systems often suffer from insufficient brightness, amplified noise, and noticeable color distortions. These degradations significantly impair the ability of intelligent detection models to accurately recognize vehicles and pedestrians.

To address such challenges, numerous illumination enhancement techniques have been developed. Liu et al. [20] highlighted the importance of systematically evaluating different enhancement strategies to understand their respective strengths and limitations in low-light conditions. Li et al. [21] introduced the Progressive Recurrent Inference Network (PRIEN), which employs a dual-attention mechanism for global feature extraction, directly enhancing low-light images in an end-to-end fashion. Lu et al. [22] proposed a dual-branch exposure fusion network that mimics the degradation process of low-light images and efficiently restores visibility by estimating illumination transfer functions across varying brightness levels. Fu et al. [23] designed LE-GAN, an unsupervised generative adversarial network that incorporates attention mechanisms and identity-invariant loss to improve enhancement quality.

Building on structural priors, Guo et al. [24] presented GLNet, which leverages gray-scale channel guidance and dense residual connections to restore fine-grained textures. Zhang et al. [25] addressed color fidelity concerns by employing a deep color consistency network that ensures natural visual appearance during enhancement. Yi et al. [26] introduced Diff-Retinex, which reconceptualizes the illumination enhancement task using a physically grounded Retinex decomposition framework, modeled as a conditional diffusion process. Hou et al. [27] extended this framework by incorporating global structure-aware diffusion dynamics and uncertainty-guided regularization, enabling more robust enhancement in visually extreme cases. In an integrated approach, Qiu et al. [28] proposed a joint optimization framework that simultaneously performs dehazing and illumination enhancement, aiming to improve detection accuracy in foggy and dimly lit traffic scenes.

This paper introduces CIDNet (Color and Intensity Decoupling Network) as the core module for image illumination enhancement and builds it on the newly proposed HVI (Horizontal/Vertical-Intensity) color space.

Traditional color spaces such as sRGB or HSV generally have problems in low-light image enhancement, such as strong brightness–color coupling and severe color distortion. The HVI color space defines a set of trainable structural mapping relationships, mapping the image from the sRGB space to the luminance and color completely decoupled HV plane and intensity channel. In the HVI space, the maximum image brightness $I_{max}$ is represented by the maximum value of the RGB channels, used to estimate the overall lighting level.(7)$$I_{max}=\underset{c\in \{R,G,B\}}{max}\left(I_{c}\right)$$

The color components are modeled for the dark areas through the introduction of density adjustment parameter $C_{k}$, thereby enhancing the color distinction in the low-light regions.(8)$$C_{k}=\frac{k}{\sqrt{sin\left(\frac{\pi I_{max}}{2}\right)+\epsilon }}$$

Based on the cosine–sine function, an orthogonal projection method is used to construct the HV plane, ensuring that the color restoration during the enhancement process has physical consistency and numerical stability.(9)$$\hat{H}=C_{k}⊙S⊙T\left(P_{\gamma }\right)⊙cos\left(2\pi P_{\gamma }\right)$$(10)$$\hat{V}=C_{k}⊙S⊙T\left(P_{\gamma }\right)⊙sin\left(2\pi P_{\gamma }\right)$$

As illustrated in Figure 2, the proposed CIDNet architecture adopts a dual-branch U-Net variant that independently processes luminance and chromatic components of low-light images. To facilitate effective interaction between these two branches, a Lighten Cross-Attention (LCA) mechanism is introduced, enabling semantic-level compensation and joint optimization across the luminance and color domains. This design significantly improves both visual clarity and structural fidelity in enhanced outputs. Experimental results demonstrate that the proposed method achieves state-of-the-art performance across multiple low-light benchmark datasets. Furthermore, in traffic surveillance scenarios, CIDNet substantially enhances the perceptual resolution of critical features such as vehicle contours and pedestrian boundaries, thereby providing a robust and high-quality image foundation for subsequent object detection tasks.

### 2.4. Defect Detection

To address the stringent requirements for accurate detection of multi-scale pedestrians and vehicles in complex traffic environments, this study proposes an enhanced detection architecture that incorporates a state-space modeling mechanism. The overall structure of the improved network, referred to as MDR-YOLO, is depicted in Figure 3.

In the Backbone stage, the model introduced the state-space-driven dual-branch perception module VSSBlock (as the basic component for feature extraction). This module consists of a local convolution branch and a state modeling path based on SS2D (State-Space 2D). SS2D models the evolution of the spatial sequence through depthwise separable convolution (DWConv) and low-rank dynamic transformation kernels, effectively capturing the global semantic dependencies of the target in long-distance traffic scenarios. By fusing the outputs of the two branches through residual connections, VSSBlock achieves the dynamic balance modeling of local edge structures and long-range dependencies, improving the discriminative robustness of the model in weak-texture and occluded target recognition. One of its core paths adopts two-dimensional state space modeling (SS2D), which can be formalized as(11)$$z_{t}=A\cdot z_{t-1}+B\cdot x_{t}$$(12)$$y_{t}=C\cdot z_{t}$$

$x_{t}$ represents the feature vector of the input image at sequence position *t*, $z_{t}$ is the hidden state vector, and *A*, *B*, and *C* are learnable state transition and mapping matrices. The output features are further fused with the local branch results to form the final output:(13)$$F_{VSS}=F_{local}+F_{SS2D}+F_{res}$$

In the feature downsampling path, this paper adopts the VisionClueMerge module to alleviate the spatial ambiguity problem that is prone to occur in the inter-layer information transmission of the traditional YOLO structure. This module rearranges the spatial information in the four quadrants of a two-dimensional space and maps it equivalently to the channel dimension. It then implements structure-aware upsampling in the subsequent convolution, effectively preserving the edge morphology and contour lines of distant traffic targets, strengthening the structural alignment ability in the deep semantic expression process. Its operation is expressed as(14)$$X_{VCM}=Concat\left(X_{00},X_{01},X_{10},X_{11}\right)$$

Here, $X_{ij}$ represents the quadrants of the input feature map. This design effectively retains high-frequency texture information such as vehicle outlines and zebra crossings, enhancing the fidelity of structural features during the deep semantic modeling process.

In the Neck stage, to further enhance the semantic consistency of the feature fusion layer, this paper proposes the multi-scale fusion enhancement module XSSBlock, which is embedded into each fusion scale path. This module builds a stacked state-space modeling path based on the *VSSBlock*. Its structure is constructed by stacking *VSSBlocks* and introducing *MLP* nonlinear transformations, and the fusion path is as follows:(15)$$F_{XSS}=MLP\left(LayerNorm\left(VSSBlock\left(X\right)\right)\right)+X$$

The DropPath path random strategy is used to enhance the generalization performance, improving the recognition accuracy of small-scale pedestrians and distant motor vehicles by the model.

In the detection head stage, this paper innovatively replaces the YOLO decoder with the RT-DETR structure’s RT-DETR Decoder module. This module integrates multi-scale feature inputs, combines the Transformer encoding mechanism and cross-layer attention fusion strategy, and realizes dynamic modeling and semantic integration of candidate regions without redundant anchor boxes. By directly utilizing global context information and multi-scale saliency responses, the RT-DETR Decoder effectively improves the accuracy of target boundary regression and classification, especially suitable for fine detection scenarios of occluded, dense, and non-rigid traffic targets. This structure further shortens the decoding path depth, reduces the information loss caused by intermediate feature transformation, and effectively improves the modeling expression ability and inference stability of the detection head.

This model, while maintaining efficient inference capabilities, significantly improves the accuracy and robustness of target recognition by introducing three key structural modules and is suitable for complex scenarios in traffic environments with challenges such as occlusion, distant small targets, and illumination changes.

## 3. Experimental Design

In the image enhancement and reconstruction task, to scientifically evaluate the performance of the model in the process of low-light image restoration, this paper selects the Structural Similarity Index (SSIM) and Peak Signal-to-Noise Ratio (PSNR) as the main evaluation indicators. SSIM aims to measure the similarity of the enhanced image and the reference image in terms of multiple dimensions such as brightness, contrast, and structure. Its value range is usually [0, 1], and the higher the value, the closer the enhanced result is to the original image in terms of structural preservation. PSNR is calculated based on the mean square error (MSE) between the image signal and the reconstruction error, and it is used to quantify the reconstruction accuracy of the enhanced image in the luminance layer. Its unit is decibels (dBs), and the higher the value, the smaller the image noise and the higher the restoration quality. The combined use of SSIM and PSNR as evaluation criteria can more comprehensively depict the actual effect of the model in the enhancement task. The specific calculation methods of the indicators are detailed in Formulas (16) and (17).(16)$$SSIM\left(x,y\right)=\frac{\left(2\mu_{x}\mu_{y}+C_{1}\right)\left(2\sigma_{xy}+C_{2}\right)}{\left(\mu_{x}^{2}+\mu_{y}^{2}+C_{1}\right)\left(\sigma_{x}^{2}+\sigma_{y}^{2}+C_{2}\right)}$$(17)$$PSNR\left(x,y\right)=10\cdot {log}_{10}\left(\frac{L^{2}}{MSE\left(x,y\right)}\right)$$(18)$$MSE\left(x,y\right)=\frac{1}{mn}\sum_{i=1}^{m}\sum_{j=1}^{n}{\left[x\left(i,j\right)-y\left(i,j\right)\right]}^{2}$$

In the task of intelligent detection of traffic vehicles and pedestrians, common performance evaluation metrics include precision, recall, average precision (AP), and mean average precision (mAP). Precision measures the proportion of correctly identified predictions among all predicted targets, reflecting the ability to control false alarms; recall indicates the proportion of true positive samples that are successfully detected by the model, which is a measure of coverage. These two metrics often have a certain contradictory relationship: a higher recall rate may come with more false alarms, thereby reducing precision, while overly pursuing high precision may lead to the omission of some targets, resulting in a decrease in recall.

To more comprehensively reflect the overall performance of the model, this paper uses AP and mAP as the main evaluation indicators. AP is the average value of precision corresponding to different recall rates for a specific category, and mAP is the mean of AP for all categories. In traffic scenarios, to balance the differences in target scales and occlusion interference, this paper adopts mAP@50–90 as the main evaluation criterion. This indicator places higher requirements on the model’s boundary positioning ability under stricter IoU thresholds and can more accurately reflect the practical value of target detection models in complex environments. The calculation methods of each indicator are detailed in Formulas (19)–(22).(19)$$Precision=\frac{TP}{TP+FP}$$(20)$$Recall=\frac{TP}{TP+FN}$$(21)$$AP=\int_{0}^{1}P\left(r\right)dr$$(22)$$mAP=\frac{1}{n+1}AP_{t}$$

### 3.1. Image Dehazing

To further verify the adaptability and image restoration effect of the proposed dehazing algorithm in real traffic monitoring scenarios, this paper selects typical urban intersection monitoring images as input samples. The ConvIR network is used for image dehazing processing, and a visual comparison analysis is conducted, as shown in Figure 4. The dehazing effect comparison on actual traffic monitoring images, from a visual perspective, reveals that there is obvious grayish-white haze covering in the original image, resulting in low overall contrast and blurred edge structures. Especially in the areas of the road and the zebra crossing, the boundary features of vehicles and pedestrians are difficult to distinguish. However, after dehazing processing by ConvIR, the image contrast is significantly improved, the color space is restored more naturally, and the edges of the zebra crossing, the body edges of motor vehicles, and the contours of cyclists are all effectively enhanced. The clarity of the structural boundaries is significantly better than that of the original image.

To verify the comprehensive performance of the adopted defogging algorithm in terms of image structure and visual quality, this paper selects three representative methods, ConvIR, ChalR, and DeHamer, and conducts quantitative evaluations on the same image dataset. The evaluation indicators include the Structural Similarity Index (SSIM) and Peak Signal-to-Noise Ratio (PSNR). The results are shown in Table 1. The SSIM index mainly measures the fidelity of image structure information and texture details, while the PSNR reflects the overall reconstruction quality and denoising level of the image. Together, they constitute the mainstream performance evaluation system for image restoration tasks.

ConvIR achieved the highest score of 0.9363, significantly outperforming ChalR and DeHamer, indicating that it can more effectively preserve the edge structure and local details of the image during the defogging process. Additionally, ConvIR also performed stably in the PSNR index, and although it was slightly lower than ChalR, it was still superior to DeHamer overall, indicating that it can effectively suppress image artifacts and noise while ensuring structural fidelity.

ConvIR has a significant advantage in structure restoration capabilities, especially suitable for image preprocessing tasks in high-frequency structure-dense areas in traffic scenarios. This method effectively improves the perception foundation of subsequent detection modules while maintaining the clarity of the defogged image, providing more robust input image quality guarantees for target detection in complex weather conditions in intelligent traffic vision systems.

### 3.2. Image Illumination Enhancement

In complex urban traffic environments, surveillance imagery captured under conditions such as nighttime, rainfall, or haze frequently suffers from reduced brightness, color distortion, and a loss of structural detail. These visual degradations pose significant challenges to intelligent detection systems, substantially impairing their ability to accurately identify and track traffic participants such as vehicles and pedestrians. As a critical component in the visual perception pipeline, the quality of illumination enhancement directly influences the robustness and precision of subsequent object detection modules.

To address these issues, this study adopts the CIDNet network, specifically designed for image enhancement in low-light scenarios. CIDNet leverages a novel HVI color space in conjunction with a decoupled reconstruction strategy, which collectively enhance the dynamic range of brightness and improve structural fidelity in challenging visual conditions.

As illustrated in Figure 5, the first image represents the original low-light input, while the second depicts the result after enhancement by CIDNet. A comparative visual analysis reveals that the original image suffers from severe brightness suppression, particularly in the road region and at the boundary between sidewalks and lanes—areas where object outlines are nearly indiscernible. Vehicle and pedestrian edges appear blurred, and critical texture information is either attenuated or lost, compromising the utility of such data for downstream visual tasks.

In contrast, the CIDNet-enhanced image exhibits uniformly improved luminance, with significantly restored detail in dark regions. Structural elements such as zebra crossings, vehicle contours, and pedestrian postures become clearly defined. The image demonstrates a substantial increase in overall contrast without introducing overexposure or noticeable color distortion. Moreover, the enhanced output maintains strong consistency in both structure and color, with no observable artifacts or unnatural textures, validating CIDNet’s effectiveness in producing high-fidelity enhancements suitable for real-world traffic scenarios.

To systematically evaluate the performance of different low-light image enhancement algorithms in terms of image structure preservation and reconstruction quality, this paper selected three representative enhancement methods: CIDNet, SG-LLIE, and Retinex, and conducted a quantitative comparative analysis on a unified low-light traffic monitoring image dataset. The evaluation indicators used include the Structural Similarity Index (SSIM) and Peak Signal-to-Noise Ratio (PSNR). The experimental results are shown in Table 2.

The results show that CIDNet achieved the highest scores in both indicators. The SSIM was 0.9233 and the PSNR was 25.94 dB. It significantly outperformed the comparison methods in terms of structural consistency and image reconstruction accuracy. SG-LLIE performed similarly in PSNR (25.90 dB), but its structural fidelity (SSIM = 0.8512) was slightly lower than that of CIDNet, indicating that it has relatively insufficient ability to preserve image details while enhancing brightness. The Retinex method, although it has a certain improvement in visual brightness, was inferior in structural restoration (SSIM = 0.6567) and reconstruction quality (PSNR = 22.41 dB), making it difficult to meet the preprocessing requirements for high-precision visual tasks.

CIDNet demonstrated excellent comprehensive performance in low-light image enhancement tasks, effectively improving the structural clarity and brightness dynamic range of the images, providing high-quality input guarantees for subsequent visual-based target detection and behavior analysis.

### 3.3. Experimental Details and Evaluation Criteria for Object Detection

The MDR-YOLO model was trained on an NVIDIA RTX 4090 GPU environment. A set of optimized key training parameters was selected to achieve efficient convergence and superior detection performance. As shown in Table 3, the input image resolution was set to 640 × 640, and the number of epochs was set to 200 to ensure that the model had sufficient time for feature learning in complex scenarios. Leveraging the powerful computational capacity of the RTX 4090, the batch size was set to 64 to enhance training stability and accelerate convergence. For optimization, the stochastic gradient descent (SGD) algorithm was employed with a momentum of 0.937 and a weight decay of 0.0005. The initial learning rate (Lr0) was set to 0.01, complemented by a warm-up and cosine annealing learning rate scheduler to achieve a better balance in the learning dynamics. Additionally, the Mosaic data augmentation strategy was enabled with a value of 1.0 to improve the model’s generalization capability in complex traffic scenarios, and the scale parameter was set to a range of 0.5–1.5 to further enhance the model’s multi-scale adaptability.

The experimental visualization results are presented in Figure 6. The object detection system was tasked with identifying three primary categories of traffic participants: car, motorbike, and pedestrians. For each detected target, the model generated bounding boxes along with associated confidence scores. In total, across four test images, the system detected 24 vehicles, 11 motorcycles, and 1 pedestrian. The distribution of detections reflects a realistic urban traffic scenario, with vehicles comprising approximately 65%, motorcycles 30%, and pedestrians 5% of the total targets.

The model exhibited particularly high stability and accuracy in vehicle detection. Among the 24 identified vehicles, 17 (70.8%) had confidence scores exceeding 0.9, and the average confidence score for all vehicle detections reached 0.91, indicating a high degree of recognition precision. For motorcycles, 8 out of 11 (72.7%) were detected with confidence scores above 0.7. The sole pedestrian was detected with a confidence score of 0.6.

The key performance indicator curves of the complete training process in the traffic scenario are shown in Figure 7. They include the loss functions during the training and validation phases, as well as the trends of model evaluation indicators over the number of iterations.

On the loss functions of training and validation, giou_loss, cls_loss, and l1_loss all decreased rapidly within the first 30 to 50 rounds and gradually stabilized. Especially in urban intersections or main roads environments, a large number of vehicles and motorcycles need to be accurately detected within a short period of time. The validation loss curve is almost completely consistent with the training loss, indicating that the model does not have significant overfitting and has good generalization ability. In terms of detection indicators, precision remained above 0.95 after the 50th round, and recall rapidly climbed in the early stage of iteration and finally reached around 0.9. In the mAP index that evaluates the overall performance of the model, mAP50 reached around 0.95 after the 60th round, and the stricter mAP50—95 also remained above 0.75. This indicates that the model has extremely strong robustness in detecting traffic participants under different IoU thresholds, especially for the quality control of the bounding box regression of pedestrians and motorcycles.

## 4. Discussion

### 4.1. Evaluation of Object Detection

To assess the individual and combined contributions of the proposed architectural components, a comprehensive ablation study was conducted on three key modules: Model A (Mamba-YOLO backbone), Model B (VSSBlock), and Model C (XSSBlock). The experimental results are summarized in Table 4.

Starting from the baseline configuration, the model achieved precision of 0.919, recall of 0.883, mAP@50 of 0.929, and mAP@50–90 of 0.707. Upon integrating the VSSBlock module (Model B), substantial improvements were observed—most notably, recall increased to 0.923 and mAP@50–90 improved to 0.756. These results underscore the effectiveness of VSSBlock in capturing multi-scale semantic features and enhancing the model’s ability to detect targets with varying sizes and occlusions.

Subsequently, the inclusion of the XSSBlock module (Model C) led to a marginal decline in precision (from 0.919 to 0.917), while recall remained relatively high at 0.896. However, a performance drop was observed in mAP@50–90, which decreased to 0.713. This degradation may be attributed to the stacked SS2D-based state-space modeling in XSSBlock, which, in highly complex visual contexts, may introduce feature redundancy and compromise localization accuracy.

Despite this, when all three modules—Mamba-YOLO + VSSBlock + XSSBlock—were combined, the model achieved optimal performance across all metrics, with precision rising to 0.945, mAP@50 reaching 0.950, and mAP@50–90 improving to 0.769. These results validate the complementarity and synergistic effect of the modules when used together.

Overall, the ablation study demonstrates that the integrated architecture strikes an effective balance between detection accuracy and robustness, confirming the benefit of combining local perception, global context modeling, and enhanced feature fusion in complex traffic detection scenarios.

In order to comprehensively verify the target detection performance of the proposed model in traffic scenarios, this paper selects the current mainstream YOLO series models, the representative Transformer structure RT-DETR18, and the lightweight baseline model Mamba-YOLO for horizontal comparison. The experimental results are shown in Table 5.

The YOLO series is renowned for high accuracy. Specifically, YOLOv12s achieves 0.948 mAP@50 and 0.759 mAP@50–95 with 9.26 M parameters, 21.5 GFLOPs and 115.7 FPS, remaining acceptable for real-time applications. RT-DETR18 delivers the best fine-grained localization (0.771 mAP@50–95), yet its 20.0 M parameters, 60.0 GFLOPs and 55.3 FPS render it impractical for real-time ITS deployment. The original lightweight Mamba-YOLO exhibits extreme efficiency (5.8 M, 13.2 GFLOPs, 120.1 FPS), yet its restricted representation limits performance under occlusion and small-object scenarios (0.747 mAP@50–95).

Our enhanced model strikes a superior accuracy–complexity balance: 0.945 precision, 0.919 recall, 0.945 mAP@50, and 0.769 mAP@50–95, rivaling RT-DETR18 while requiring only 9.07 M parameters, 17.2 GFLOPs and 117.7 FPS. This substantial reduction in memory and computation, coupled with maintained precision, endows the model with exceptional deployment friendliness. Although the lightweight trade-off marginally compromises localization, the holistic competitiveness remains compelling.

Thanks to its compact architecture and robust generalization, the model performs reliably across diverse scenes—urban intersections, non-motorized lanes, and nighttime surveillance—accurately detecting vehicles and pedestrians under challenging conditions, thus seamlessly aligning with modern ITS infrastructure requirements.

To rigorously assess the proposed model’s efficacy under multi-scale scenarios—particularly for small-object detection—we concentrate on two representative categories, pedestrian and motorcycle, and benchmark against three top-performing lightweight YOLO variants (YOLOv5s, YOLOv8s, and YOLOv12s). Quantitative comparison reveals that our method consistently delivers superior fine-grained localization and scale robustness.

As reported in Table 6, the overall accuracy remains moderate, owing to the limited presence of motorcycles and pedestrians in the dataset. Nevertheless, our model attains the best AP@50–95 of 0.623 for motorcycles and 0.738 for pedestrians, thereby confirming its leading small-object detection capability under data-scarce conditions.

To comprehensively assess the deployment viability of our lightweight model, we benchmark both accuracy and real-time performance on an RTX 4090 desktop (NVIDIA Corporation, Santa Clara, California, USA) and a Raspberry (Raspberry Pi Foundation, Cambridge, United Kingdom) Pi 4B emulator (4× Cortex-A72 @ 1.5 GHz). As shown in Figure 8, on the RTX 4090 our model achieves the highest FPS among YOLOv5n and YOLOv7-tiny while delivering the best mAP. On the emulated embedded platform, although the FPS is slightly lower than that of YOLOv5n and YOLOv7-tiny, it still sustains real-time inference with superior accuracy.

### 4.2. Image Processing Effect Evaluation

In traffic visual perception scenarios, raw images are frequently affected by environmental factors such as low illumination and haze, leading to reduced contrast, blurred edges, and poorly defined target structures. To systematically evaluate the practical impact of the proposed image preprocessing modules on detection performance, a controlled comparative experiment was designed in which the input variable was the presence or absence of image enhancement. This approach enables a quantitative assessment of the contribution and necessity of image defogging and illumination enhancement within the overall detection pipeline.

Two experimental conditions were constructed:Detection results using images before and after defogging.Detection results using images before and after illumination enhancement.

All experiments were conducted using the same detection network architecture and hyperparameter settings to ensure a fair comparison. The primary evaluation metrics were mAP@50 and mAP@50–90, providing insight into both coarse and fine-grained detection accuracy.

As shown in Figure 9, the left image illustrates detection results under low-light conditions, while the right image displays results after illumination enhancement. In the original (unprocessed) image, severe underexposure leads to substantial information loss in dark regions. As a result, object boundaries appear indistinct, especially in the upper portion of the frame, where the model struggles to identify distant vehicles and pedestrians. Confidence scores for near-field vehicle detections are relatively low, with values of 0.88 and 0.85, and the confidence for a motorcycle at the bottom of the frame is only 0.78, reflecting the model’s limited perceptual capability in poorly illuminated environments.

In contrast, the enhanced image shows significant improvements in overall brightness, contrast, and structural visibility. The contours of vehicles are clearly defined, and target integrity is preserved. Detection coverage is notably expanded, extending from the mid-ground to distant areas of the image. Nearly all vehicles are successfully identified, with the front three vehicles receiving confidence scores of 0.90, 0.89, and 0.94, respectively. Even distant vehicles in the upper lanes are correctly detected, many with confidence values exceeding 0.90. Furthermore, the confidence score for the previously low-confidence motorcycle increases to 0.84, indicating enhanced model awareness across the scene.

These results confirm that image enhancement preprocessing effectively mitigates visual degradation caused by low-light conditions. It substantially improves the detectability, confidence distribution, and spatial coverage of targets, thereby enhancing the overall detection accuracy. The integration of such preprocessing modules offers strong practical support for reliable object detection in visually complex and dynamically illuminated traffic environments.

As shown in Figure 10, the left image illustrates the detection outcome on the original, unprocessed input, whereas the right image presents the results after applying both defogging and illumination enhancement. In the original image, the presence of haze and low overall brightness leads to reduced visual clarity and blurred target boundaries, particularly for mid- and long-range objects. While most targets remain detectable, confidence scores show moderate degradation. Specifically, the detection confidence for the black vehicle on the left is 0.92, the motorcycle below registers 0.90, and the bicycle on the right—located in the non-motor vehicle lane—achieves 0.88. However, the image lacks clear visual stratification and sufficient contrast, which may compromise boundary localization accuracy.

In contrast, the right image—after preprocessing—demonstrates enhanced visual quality. The overall brightness is elevated, haze interference is effectively removed, and target textures and edges are significantly clearer. The detection confidence for most targets remains stable or shows slight improvement: the white vehicle in the main lane maintains a confidence of 0.94, the motorcycle registers 0.91, and the bicycle’s confidence increases to 0.90. The refined image structure facilitates more compact and accurate bounding box generation, improving edge delineation and spatial consistency.

These results collectively highlight the practical benefits of image enhancement. The preprocessing pipeline substantially improves overall visibility and fine-grained detail representation, leading to more stable and accurate detection results. Enhanced edge definition and texture recovery support superior structural perception and spatial localization, especially in multi-target traffic scenes.

By comparing the detection performance before and after illumination enhancement and dehazing, it becomes evident that image preprocessing plays a pivotal role in boosting detection system effectiveness. Illumination enhancement significantly increases the perceptibility and textural clarity of dark regions, enhancing the model’s sensitivity to low-light targets and increasing both the number and confidence of detections. Simultaneously, dehazing effectively mitigates structural degradation caused by blurring and occlusion, thereby improving the edge clarity, local detail expression, and detection stability.

Overall, the dual-stage image restoration process—comprising both dehazing and low-light enhancement—successfully addresses the perceptual degradation challenges in complex traffic environments. It lays a solid foundation for downstream detection tasks by delivering high-quality input, ultimately reinforcing the robustness, precision, and reliability of intelligent traffic perception systems.

To rigorously evaluate the contribution of the proposed image enhancement module to the generalization of the object detector under adverse weather, we conducted an ablation study on the Cityscapes-Adverse dataset [29] (2500 images: 20% hazy, 20% low-light, 60% normal). As listed in Table 7, the baseline without any enhancement achieves 0.715 mAP@50–90. The haze-removal branch alone raises this to 0.742 (+3.7 pp), while the illumination–enhancement branch yields a larger gain to 0.791 (+7.6 pp). Cascading the two modules further elevates the metric to 0.811, amounting to a total improvement of 9.6 pp, which substantiates the complementary effects of dehazing and illumination correction in the feature space and quantitatively confirms the efficacy of the proposed enhancement strategy.

## 5. Conclusions

This paper presents a robust vision-only detection framework tailored for vehicle and pedestrian recognition in complex urban traffic. By integrating a multi-stage image-enhancement pipeline with a lightweight yet accurate detector (MDR-YOLO), the system maintains high performance under adverse weather, low illumination, and severe occlusion. Extensive experiments demonstrate that (i) the proposed preprocessing module yields consistent and significant improvements in detection metrics under degraded visual conditions and (ii) MDR-YOLO attains state-of-the-art accuracy–efficiency trade-offs, making it well-suited for embedded traffic-monitoring platforms.

Limitations and future work include generalization to atypical scenes such as construction zones and the incorporation of complementary modalities (millimeter-wave radar, LiDAR, or infrared) to evolve toward a truly multimodal, all-weather perception system.

## Figures and Tables

**Figure 1 sensors-25-05014-f001:**
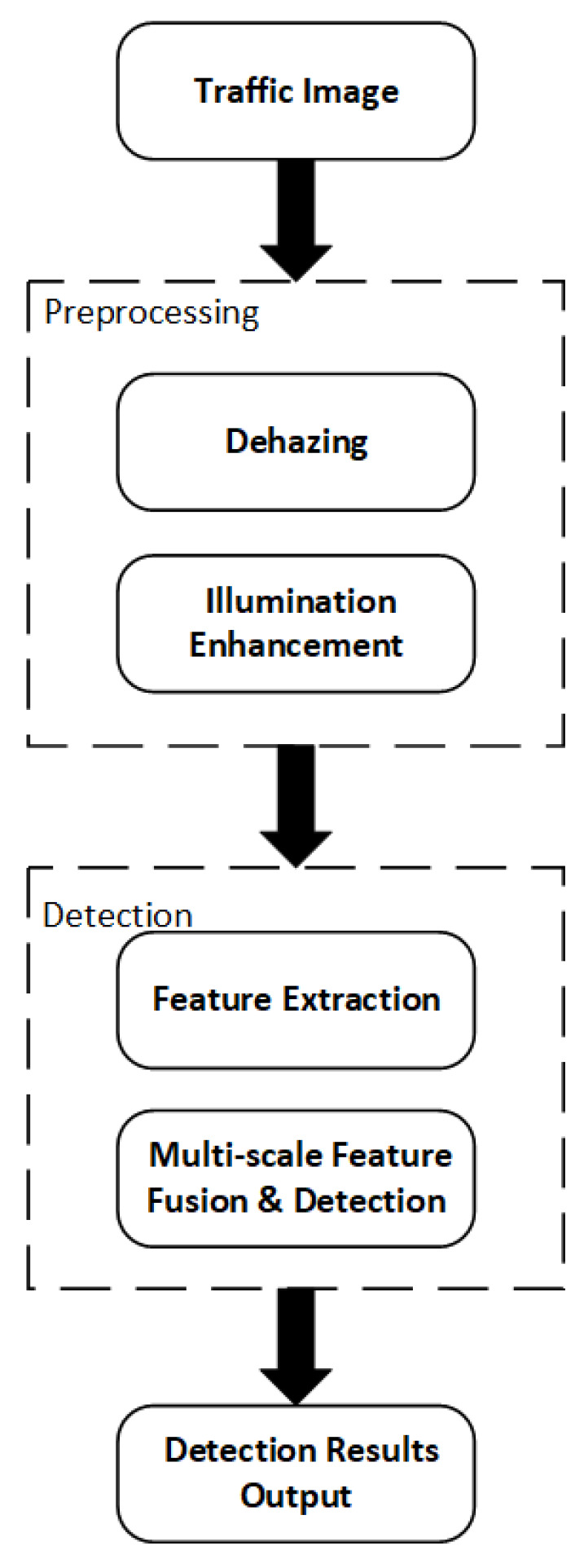
Overall flowchart of the traffic detection system.

**Figure 2 sensors-25-05014-f002:**
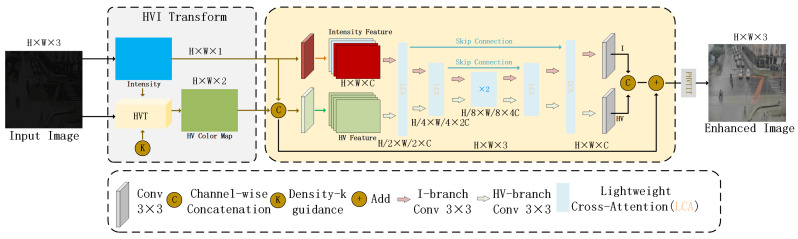
CIDNet network structure diagram.

**Figure 3 sensors-25-05014-f003:**
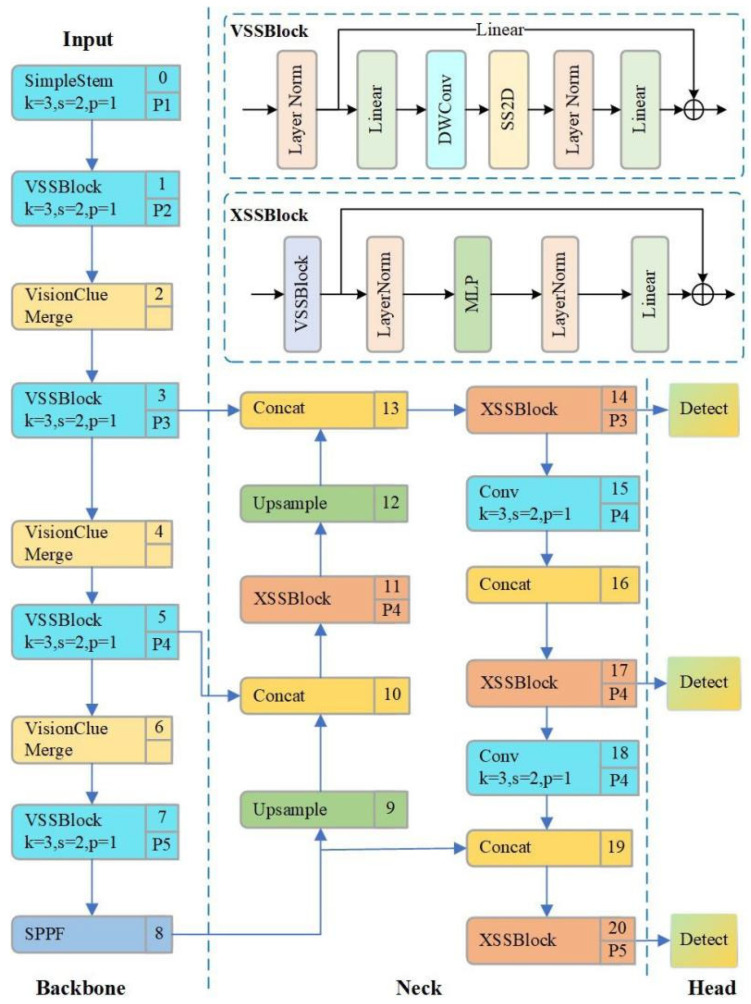
MDR-YOLO network architecture diagram.

**Figure 4 sensors-25-05014-f004:**
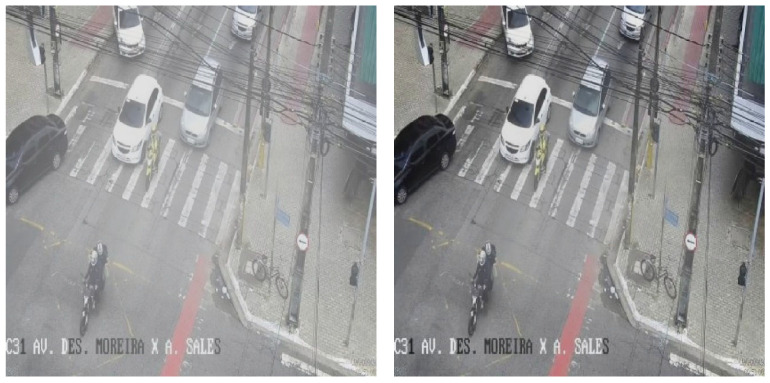
Comparison chart of dehazing effects based on ConvIR.

**Figure 5 sensors-25-05014-f005:**
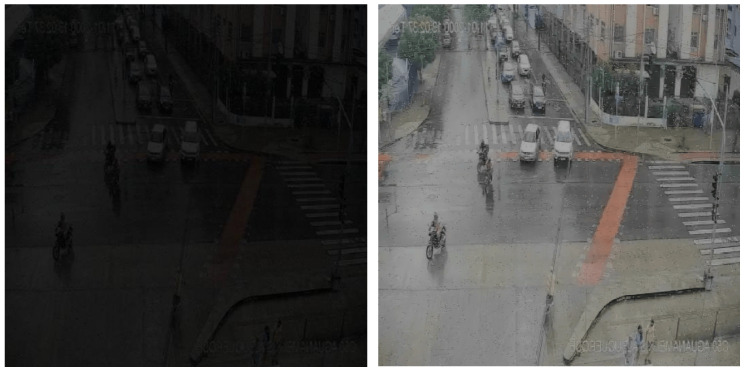
Comparison chart of illumination enhancement effect Bbased on CIDNet.

**Figure 6 sensors-25-05014-f006:**
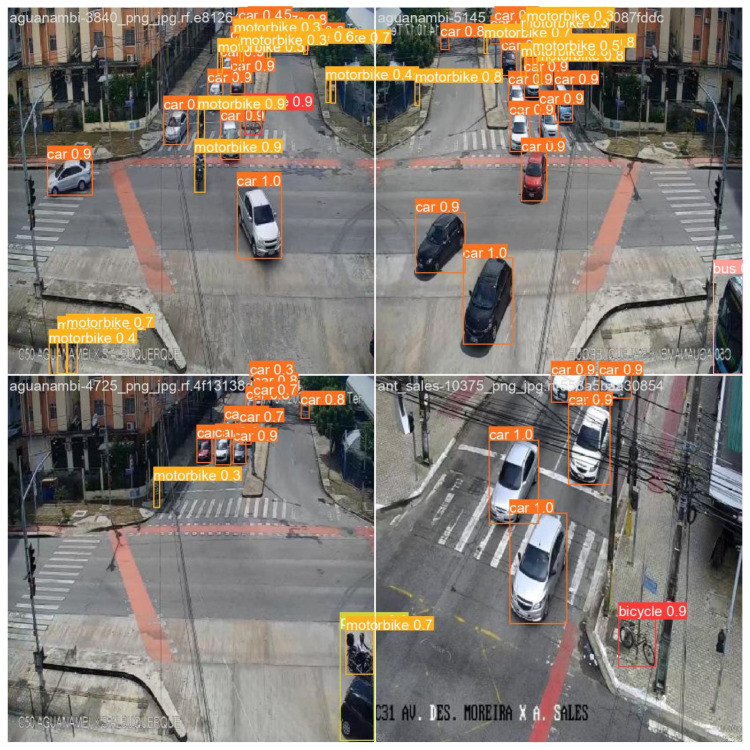
Detection results based on the MDR-YOLO method.

**Figure 7 sensors-25-05014-f007:**
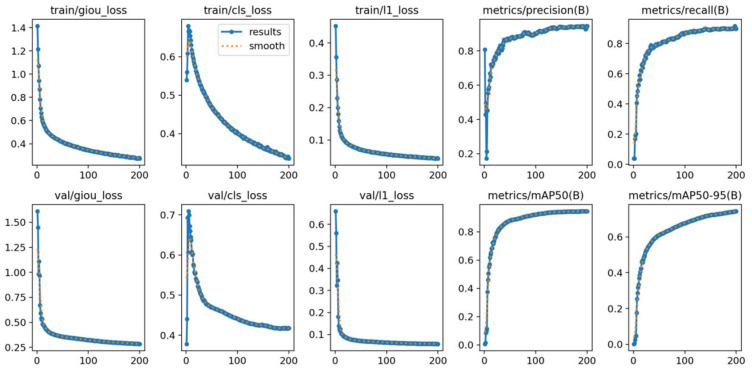
Training process of MDR-YOLO.

**Figure 8 sensors-25-05014-f008:**
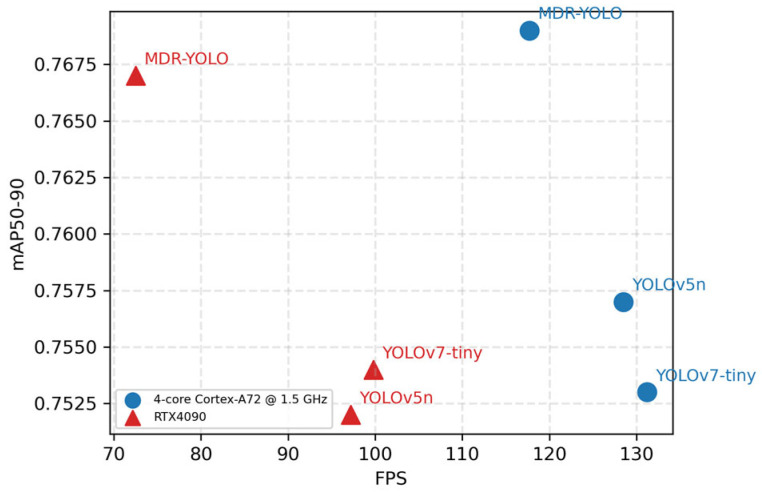
Speed–accuracy trade-off of different models across hardware platforms.

**Figure 9 sensors-25-05014-f009:**
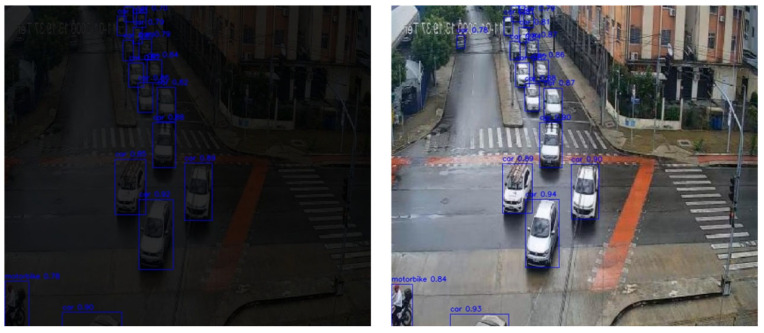
Detection comparison chart based on illumination enhancement.

**Figure 10 sensors-25-05014-f010:**
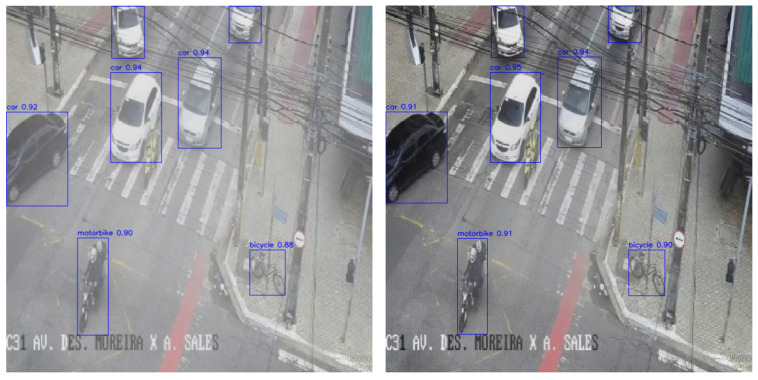
Comparison chart of image dehazing detection.

**Table 1 sensors-25-05014-t001:** Comparison of evaluation indicators for the fog removal algorithm.

Task	Arithmetic	SSIM	PSNR (dB)
	ConvIR	0.9363	19.29
Dehaze	ChaIR	0.9085	19.31
	DeHamer	0.8979	17.19

**Table 2 sensors-25-05014-t002:** Comparison of Evaluation Indicators for Illumination Enhancement Algorithm.

Task	Algorithm	SSIM	PSNR (dB)
Low-light Image Enhancement	CIDNet	0.9233	25.94
	SG-LLIE	0.8512	25.90
	Retinex	0.6567	22.41

**Table 3 sensors-25-05014-t003:** Configuration of MDR-YOLO Training Parameters.

Parameter	Value
Image size	640 × 640
Epochs	200
Batch size	64
Lr0	0.01
Optimizer	SGD
Mosaic	1.0
Scale	0.5–1.5

**Table 4 sensors-25-05014-t004:** Abolition Study of Key Modules.

Model A	Model B	Model C	P	R	Map50	Map50–90
√			0.919	0.883	0.929	0.707
√	√		0.921	0.923	0.943	0.756
√		√	0.917	0.896	0.937	0.713
√	√	√	0.945	0.919	0.945	0.769

**Table 5 sensors-25-05014-t005:** Ablation experiments comparing the improved model with various baseline models.

Model	P	R	Map50	Map50–90	Param (M)	Flops (G)	FPS
Yolov5s	0.927	0.917	0.941	0.749	7.50	16.5	120.7
Yolov6s	0.918	0.912	0.936	0.741	4.3	12.3	138.2
Yolov8s	0.930	0.918	0.943	0.750	11.2	28.6	103.5
Yolov9s	0.933	0.916	0.945	0.754	11.3	29.0	100.7
Yolov10s	0.936	0.921	0.946	0.757	11.0	30.3	101.3
Yolov11s	0.937	0.918	0.947	0.755	9.4	21.5	116.2
Yolov12s	0.939	0.920	0.948	0.759	9.26	21.5	115.7
RT-DETR18	0.935	0.920	0.949	0.771	20.0	60.0	55.3
Mamba-Yolo	0.919	0.883	0.929	0.747	5.8	13.2	129.1
Ours	0.945	0.919	0.945	0.769	9.07	17.2	117.7

**Table 6 sensors-25-05014-t006:** Small-object detection performance across models.

Model		Yolov5s	Yolov8s	Yolov12s	Ours
evaluation metric		AP@50–95			
small target object	Motorbike	0.579	0.610	0.583	0.623
	Pedestrian	0.665	0.716	0.681	0.738

**Table 7 sensors-25-05014-t007:** Quantitative Performance of Image Preprocessing Strategies.

Task	P	R	Map50	mAP@50–90
no enhancement	0.742	0.627	0.776	0.715
dehazing	0.785	0.735	0.793	0.742
illumination enhancement	0.778	0.738	0.791	0.737
combined enhancement	0.823	0.749	0.811	0.751

## Data Availability

All data used in this study are publicly available: traffic-detection images were obtained from the “Traffic Detection Project” dataset on Kaggle (no DOI assigned), and urban-scene images were downloaded from the Cityscapes-Adverse dataset on GitHub (DOI: 10.1109/ACCESS.2025.3537981); both are accessible free of charge.

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
