# Peer review of "A Lightweight Multi-Stage Visual Detection Approach for Complex Traffic Scenes"

_sensors, 2025, doi:10.3390/s25165014_

Round 1

Reviewer 1 Report

Comments and Suggestions for Authors

This manuscript proposes a lightweight multi-stage visual detection approach for complex traffic scenes. It is well-written and easy to read and understand. The work has certain scientific contributions. I only have some minor suggestions:

  1. If possible, it is recommended to use university email address as the corresponding author’s email address.
  2. Please try to avoid the use of abbreviations such as HVI, MDR, YOLO in the abstract as these might cause confusion when there is a lack of context.
  3. The authors should give the full name of an abbreviation (e.g., DAWN, YOLO, RT-DETER) if it first occurs in the manuscript. In addition, if a term will not be used after its first occurrence, there is no need to give its abbreviation.
  4. Please summarize the research gaps and clarify your research motivations after reviewing the literature in the introduction section. Besides, the organization of the remaining parts of the manuscript should be described at the end of the introduction section.
  5. No text cites Figure 1. The figure is possibly not needed as it conveys nothing.
  6. In many equations, such as Eqs. (1), (3), (4), (5), (10) to (21), variables should be displayed in italic.
  7. Please clarify the novelties of the proposed approach when describing each theoretical component in section 2.
  8. Please add a description of the details of data preparation, experiment machine configuration, related parameters, as well as other settings to facilitate future reproducing the experiments.
  9. In the conclusion section, there is no need to emphasize the contributions again as this has been done in the introduction section. Instead, the authors should summarize the research findings, as well as possible limitations of the research and the approach and how do these imply future research in this area.
  10. The text size in figures 2, 3, 4, 8 is a little bit small. If possible, please enlarge the text is these figures.

Reviewer 2 Report

Comments and Suggestions for Authors
  1. Experimental design and dataset information need clarification:

The manuscript lacks a clear description of the traffic surveillance dataset used in the experiments. Key details such as dataset name, scale, and scene distribution (e.g., the proportion of foggy, low-light, and occluded samples), as well as annotation categories, are not provided. If a publicly available dataset is used (e.g., DAWN or the adverse weather subset of Cityscapes), appropriate citations should be included. If a proprietary dataset is used, further details regarding data collection—such as acquisition equipment, time, and location—should be added.

In the performance comparison, RT-DETR achieves a slightly higher mAP@50–90 (0.771) than the proposed model (0.769), but with significantly more parameters and computational overhead. To support the claim of superior deployment adaptability, real-time performance metrics (e.g., FPS) should be included. Additionally, the slight decrease in accuracy should be explained—possibly due to the trade-off from the lightweight design, which may sacrifice fine-grained localization precision.

In the ablation study, the combination “Mamba-YOLO + VSSBlock + XSSBlock” achieves an mAP@50–90 of 0.743, while the final model (Ours) reaches 0.769. However, the source of this improvement is not explained. If additional modules (e.g., VisionClueMerge) contributed to the performance gain, individual validation experiments should be included to isolate their effect.

  1. Practical deployment performance of the lightweight model is under-explored:

The paper emphasizes the advantages of low parameter count and low computational complexity, supported by FLOPs and Params metrics. However, it lacks empirical or simulated validation of inference speed, latency, and power consumption on real-world deployment platforms such as embedded systems or low-power devices. It is recommended to provide a trade-off comparison (e.g., speed–accuracy curve) with other lightweight models such as YOLOv5n and YOLOX-Tiny to demonstrate practical applicability.

  1. Result analysis needs greater depth:

The contribution of image preprocessing to detection performance is only discussed qualitatively (Figures 9 and 10), without supporting quantitative data. To substantiate the effectiveness of preprocessing, a comparative analysis should be added across four scenarios: (1) no enhancement, (2) dehazing only, (3) illumination enhancement only, and (4) combined enhancement. Metrics such as mAP@50–90, precision, and recall should be reported for each case.

Additionally, the paper lacks analysis of detection performance on small objects (e.g., distant pedestrians or motorcycles). It is recommended to report AP values specifically for small targets to validate the effectiveness of MDR-YOLO in multi-scale object detection, ideally in comparison with models like YOLOv12s.

Round 2

Reviewer 2 Report

Comments and Suggestions for Authors

The revisions are satisfactory and adequately address the raised issues. I recommend publication.